# Recovery of the lumbopelvic movement and muscle recruitment patterns using motor control exercise program in people with chronic nonspecific low back pain: A prospective study

**Sharon M. H. Tsang**[1,2☺]*, **Grace P. Y. Szeto**[3☺], **Angelina K. C. Yeung**[2‡], **Eva Y. W. Chun**[2‡], **Caroline N. C. Wong**[2‡], **Edwin C. M. Wu**[2‡], **Raymond Y. W. Lee**[4‡]

1 Department of Rehabilitation Sciences, The Hong Kong Polytechnic University, Hong Kong SAR, China, 2 Department of Physiotherapy, Prince of Wales Hospital, Hong Kong SAR, China, 3 School of Medical and Health Science, Tung Wah College, Hong Kong SAR, China, 4 School of Technology, University of Portsmouth, Portsmouth, United Kingdom

☺ These authors contributed equally to this work.
‡ These authors also contributed equally to this work
* Sharon.Tsang@polyu.edu.hk

## Abstract

This study aims to investigate the dysfunction and recovery of the lumbopelvic movement and motor control of people with chronic nonspecific low back pain after a structured rehabilitation which emphasizes on re-education and training of movement and motor control. The lumbopelvic movement and motor control pattern of 30 adults (15 with chronic low back pain, 15 healthy controls) were assessed using 3D motion and electromyographic analysis during the repeated forward bending test, in additional to the clinical outcome measures. Regional kinematics and muscle recruitment pattern of the symptomatic group was analysed before and after the 6-week rehabilitation, and compared to healthy controls. Significant improvement in back pain, functional capacity and self-efficacy of the symptomatic group was found after the rehabilitation. Patients with chronic nonspecific low back pain were capable to recover to a comparable level of the healthy controls in terms of their lumbopelvic movement and motor control pattern upon completion of a 6-week rehabilitation program, despite their dysfunction displayed at baseline. Phase specific motor control reorganization in which more profound and positive changes shown during the flexion phase. Our findings indicate that the recovery of the movement and motor control pattern in patients with chronic low back pain achieved to a comparable level of the healthy able-bodies. The improvement of both the physical outcome measures suggest that specific rehabilitation program which emphasizes on optimizing motor control during movements would help promoting the functional recovery of this specific low back pain subgroup.

**Data Availability Statement:** All relevant data are within the paper and its Supporting information files.

**Funding:** ST received the Departmental Research Fund, Department of Rehabilitation Sciences, The Hong Kong Polytechnic University, Hong Kong (Reference no.: 1-ZE4D). The funders had no role in study design, data collection and analysis, decision to publish, or preparation of the manuscript.

**Competing interests:** The authors have declared that no competing interests exist.

# Introduction

Chronic low back pain (LBP) is the leading cause of disability and it is associated with serious socioeconomic burden globally [1] Various predisposing factors have been studied in previous research for their relation to LBP. Some evidences suggested that forward bending contributes to the increase in the risk of development and/or aggravation of LBP [2–4]. Repetitive exposures to shear forces on the intervertebral discs and ligaments of the lumbar spine constitutes one of the highest risks of back injury in workers who perform frequent bending tasks [5–7]. Forward bending performed in standing is simple and routine part of the clinical assessment to evaluate the spinal movements and motor control in people with LBP [8–10]. However, conflicting results were found in the regional contribution of the lumbo-pelvic regions during bending task in patients with LBP. Esola et al. [11] reported that people with LBP moved with a similar degree of mobility at both the lumbar spine and hip joint as healthy individuals, when bending forward in standing. In contrast, Porter et al. [12] found that individuals with LBP moved with significantly decreased lumbar mobility in flexion, compared to an asymptomatic individual. These inconsistent findings of lumbar and/or hip mobility reported could partly be related to heterogeneous causes of LBP, which include motor control and/or mobility dysfunction of the lumbo-pelvic region [10, 13].

Time trajectory at which the forward bending movement initiates and coordinates during the movement cycle had also be investigated before. Flexion of the trunk normally initiates at the lumbar spine when one performs forward bending in standing [14]. It has been reported that a greater contribution of motion at the lumbar spine relative to the hip joint during the early phase of trunk flexion, with a ratio of approximately 2:1 for the two respective regions found among pain-free individuals [11]. During the late phase of bending, hip motion becomes predominant and the ratio of lumbar-to-hip motion drops to 2:5. However, inconsistent findings of the mobility restriction between individuals with and without LBP have been reported in previous studies [15–17]. Tsang et al. [15] found that it was not the mobility but the impaired movement velocity as well as coordination between the lumbar spine and hip joint that showed most apparent differences between the healthy and symptomatic groups. Furthermore, weak correlation has been found consistently between the lumbar mobility and functional disability, for which it further relegates the relative contribution of mobility dysfunction to the functional capacity in chronic LBP [18]. Meanwhile, emerging evidence indicates the importance and specificity of assessing the limits of stability and movement velocity in chronic LBP because these variables were highly associated with the degree of functional disability in chronic LBP [19]. These findings reinforce the importance to assess and optimize the movement strategy for those with underlying motor control dysfunction [20, 21]. It has been showed that exercise therapy is effective for pain relief and functional recovery for chronic LBP. However, the effects are small and there is no evidence to suggest that one specific type of exercise is distinctly superior compared to others [22, 23]. There is emerging evidence to indicate there are greater benefits associated with strength/endurance and coordination/stabilisation exercise programs compared to aerobic exercise program [24]. Laird et al. conducted a systematic review examining the effects of movement-based interventions in the lumbo-pelvic kinematics for people with chronic LBP [25]. Movement-based interventions refer to structured programs that are formulated by movements and exercises at specific directions based on the movement-based classification, for optimising the quality of movement execution and producing pain relief for individuals with musculoskeletal dysfunction [25–27]. However, they found that movement-based interventions were infrequently effective for changing the clinically recognizable movement patterns. Furthermore, changes in movement patterns failed to show significant and consistent association with the improvement in pain or functional limitation.

The purposes of this study were to compare the lumbo-pelvic movements during forward bending in individuals with and without chronic LBP; and to investigate the modification of the movement and motor control for those with chronic LBP after completing a physiotherapist supervised rehabilitation program which emphasizes on movement quality and control re-education and training.

## Materials and methods

### Participants

Thirty adults (15 with LBP [8 females and 7 males] and 15 age and gender-matched asymptomatic participants) were recruited from the Department of Physiotherapy of Prince of Wales Hospital, Hong Kong (Table 1). Inclusion criteria for LBP group included nonspecific back pain experienced between L1 and the gluteal fold without radiation to their lower limbs and lasted > 3 months. Fifteen adults who were asymptomatic for the last 12 months were recruited as the healthy controls with their age- and gender-matched with the LBP group. Exclusion criteria applied to both groups, which included the presence of pain, pathology or deformity of the hip joint; history of operation, known orthopaedic or neurological conditions of the spine and/or hip; or vestibular dysfunction. Ethical approvals for this study were obtained from the Ethics Committee of the Hong Kong Polytechnic University and Cluster Research Ethics Committee of Prince of Wales Hospital. All participants provided their written informed consents before the study commenced. The individual in this manuscript has given written informed consent (as outlined in PLOS consent form) to publish these case details. All procedures of this study were carried out in accordance with Declarations of Helsinki 2018.

### Procedures

**Clinical outcome assessments.** Participants in LBP group were asked to complete a set of five questionnaires or scales to screen for their prognostic indicators, and to quantify their level of fear avoidance to movement, functional disability associated with back pain and self-efficacy respectively. The validated tools included 1) STarT Back Tool (SBST) and 2) Tampa

**Table 1. Demographics and clinical outcomes presented with the mean ± standard deviation.**

|  | Healthy (n = 15, 8F:7M) | LBP (n = 15, 8F:7M) |  | p value |
|---|---|---|---|---|
| Age (years) | 33.7 ± 4.4 | 34.1 ± 4.8 |  | 0.83 |
| Weight (kg) | 56.31 ± 12.74 | 62.89 ± 17.21 |  | 0.262 |
| Height (cm) | 164.67 ± 9.35 | 161.38 ± 9.76 |  | 0.852 |
| BMI (kg/m$^2$) | 20.53 ± 2.68 | 23.23 ± 4.2 |  | 0.073 |
| Clinical outcomes (for LBP group only) |  | Pre | Post | p value |
| VAS | -- | 2.75 ± 0.62 | 0.17 ± 0.39 | <0.001* |
| RMQ | -- | 6.27 ± 4.29 | 4.36 ± 2.42 | 0.412 |
| SBTB (TOTAL) | -- | 3.18 ± 1.89 | 1.91 ± 1.87 | 0.214 |
| SBTB (SUB) | -- | 1.73 ± 1.27 | 1.27 ± 1.42 | 0.474 |
| PSFS | -- | 3.92 ± 1.64 | 7.51 ± 1.39 | <0.001* |
| TSK | -- | 44 ± 4.86 | 40.91 ± 4.59 | 0.053 |
| PSEQ | -- | 41.36 ± 7.94 | 47.27 ± 9.56 | 0.046* |
| Hamstrings flexibility | 21.68 (10.66) | 23.64 ± 5.97 | 25.03 ± 5.90 | 0.102 |

BMI, Body Mass Index; NPRS 0–10, Numeric Pain Rating Scale 0–10; PSFS, Patient Specific Functional Scale; PSEQ: Pain Self Efficacy Questionnaire; RMQ, Roland Morris Questionnaire; SBTB, STarT Back Tool; TSK: Tampa Scale of Kinesiophobia; VAS, Visual Analogue Scale.

Scale of Kinesiophobia (TSK) [28, 29]; 3) Roland Morris Disability Questionnaire (RMDQ) and 4) Patient Specific Functional Scale (PSFS) [30, 31]; and 5) Pain Self Efficacy Questionnaire (PSEQ) [32] before and after the rehabilitation program (S1 File).

**Physical outcomes assessments.** All participants were instructed to perform the repeated forward bending while standing by reaching down as far as they could with their hands while keeping their elbows and knees fully extended. No attempt was made to correct the movement for examining the natural movement pattern adopted by individual participants. For participants who could not reach the floor, the lowest point that was reached by his middle finger was defined as the end point by placing a foam on the ground for task standardization for individual participants. Participants were asked to perform the bending task for 7 times consecutively at their self-preferred speed. Each bending movement cycle is defined as "bending from the standardized standing position until reaching the end point set and recovery to the same standing position" (Fig 1). Participants were allowed to practise the bending task 3 to 5 times before the actual test. All participants were asked to refrain from performing heavy physical activity of their lower limbs in the 48-hour period prior to the assessment and reassessment sessions.

**Surface electromyography and kinematics of the lumbopelvic region.** Surface electromyography (EMG, MyoMuscle, Noraxon, USA) was used to investigate the activation pattern of 7 pairs of lumbo-pelvic muscles (3 ventral and 4 dorsal muscle pairs) during the repeated forward bending task, at a sampling frequency of 1024Hz. The seven muscles examined were: left and right rectus abdominus (RA), internal oblique (IO), tensor fascia latae (TFL), lumbar erector spinae (LES), lumbar multifidus (MUL), gluteus maximus (Gmax) and biceps femoris (BF). These muscles were chosen to be studied because of their functional roles as the prime movers and/or stabilisers of the lumbopelvic region during flexion (standing to end of bending) and extension (end of bending to standing) phases of the task [33, 34]. Standardized skin preparation was applied to achieve the skin impedance <10 kΩ for EMG electrode placement [35]. Disposable bipolar Ag-AgCl surface electrodes (10mm Ø) were used with inter-electrode

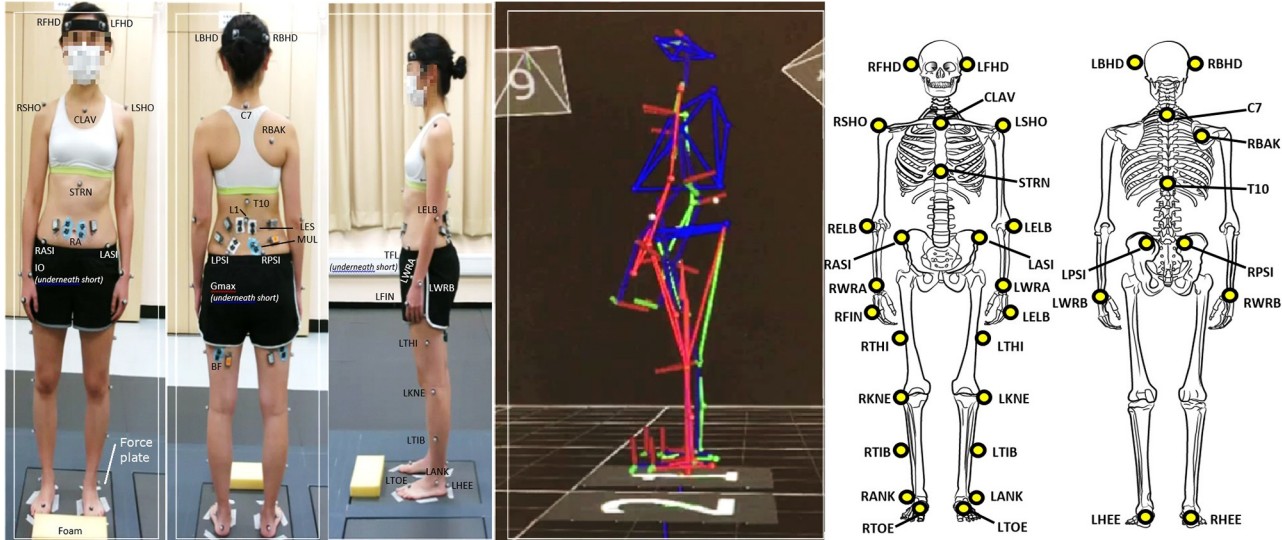

**Fig 1. The experimental setup for measuring the kinematics and electromyography during repeated forward bending test.** Reflective markers were placed for the Plug-in Gait full body model (FHD: BHD: SHO: CLAV: clavicle; STRN: sternum; ASI: anterior superior iliac spine; C7: 7th cervical spinous process; BAK: T10: 10th thoracic spinous process; PSI: posterior superior iliac spine; ELB: elbow; WRA: wrist over radial styloid process; WRB: wrist over ulnar styloid process; FIN: head of 2nd metacarpal; THI: thigh; KNE: knee; TIB: tibia; ANK: lateral malleolus; TOE: 2nd metatarsal; HEE: posterior calcaneus; with additional marker over L1: 1st lumbar spinous process. EMG over RA: rectus abdominus; IO: internal oblique; TFL: tensor fascia latae; LES: lumbar erector spinae; MUL: lumbar multifidus; Gmax: gluteus maximus; BF: biceps femoris).

distance of 20mm. EMG electrodes were placed on the muscles according to the recommendations of SENIAM and previous studies [36–38]. RA: 1cm above umbilicus and 2cm lateral to the midline [39]; IO: 1cm medial to the anterior superior iliac spine (ASIS) and below a line connecting the left and right ASISs [37]; TFL: proximal 1/6 along the line from ASIS to lateral femoral condyle [39]; LES: between L1 and L2 spinous process and midway of the muscle belly [39]; MUL: parallel to the line between posterior superior iliac spine (PSIS) and L1-L2 interspinous space at L5 level [40]; Gmax: midway on the line between sacral vertebrae and greater trochanter [39]; BF: midway on the line between ischial tuberosity and lateral epicondyle of the tibia [39]. Normalization of the EMG amplitude to the percentage of maximal isometric voluntary contraction (% MVC) value of the corresponding muscles was compared between muscles, groups, and assessment intervals. MVC procedures for the respective muscles were conducted according to the procedures reported previously [41–43].

Kinematics of the spine and lower extremities was recorded using the optical motion capturing system equipped with 10 cameras (Vicon, Oxford, UK). A total of 35 markers were positioned according to the Plug-in Gait (version 2.0) full body model and an additional one placed over L1 spinous process, Fig 1). Placement of the reflective markers included FHD: forehead; BHD: behind head; SHO: posterior scapula; CLAV: clavicle; STRN: sternum; ASI: anterior superior iliac spine; C7: 7th cervical spinous process; BAK: T10: 10th thoracic spinous process; PSI: posterior superior iliac spine; ELB: elbow; WRA: wrist over radial styloid process; WRB: wrist over ulnar styloid process; FIN: head of 2nd metacarpal; THI: thigh; KNE: knee; TIB: tibia; ANK: lateral malleolus; TOE: 2nd metatarsal; HEE: posterior calcaneus; with additional marker over L1: 1st lumbar spinous process. Kinematics was acquired at the sampling frequency of 200Hz. Calibration was conducted with the participants adopted the natural standing position with their feet standardized at the shoulder-width apart and eyesight maintained at a target placed 5m in front of them at eye level. In addition, the excursion of the Centre of Pressure (COP) of the feet of the participants in the antero-posterior and medio-lateral directions during the execution of the repeated forward bending test was recorded by the force plate (PF5000, AMTI, Massachusetts, USA) connected to the optical motion capturing system [44].

## 6-week structured rehabilitation program

After the baseline assessment, all participants in LBP group attended the supervised rehabilitation program at the outpatient clinic of Department of Physiotherapy of Prince of Wales Hospital, twice/week for 6 consecutive weeks. The program was supervised by a physiotherapist who had >15 years of experience in the musculoskeletal rehabilitation field. Each session lasted for 45 minutes in a ratio of 1 physiotherapist to 5 patients. Each session consisted of 5-minute warm up and 5-minute cool down (stretching of the back extensors and thigh muscles) before and after a 35-minute core muscles and movement control exercises using therapeutic balls and therabands. Emphasis of optimal posture and control of the movement at the respective spine regions with individual guidance for correcting the faulty movement pattern. The ability and accuracy of the activation of the anterior core muscles (namely transversus abduminus and internal oblique) of individual participants was assessed by the responsible physiotherapist in the first session using standardised palpation procedures [45], and corrected during the practice sessions of the exercise program. Core stability and control exercises included hamstring roll-ins, hip raise, ball crunch, roll-out, sitting marching and wall squat with the use of a therapeutic ball with each exercise repeated for ten times [46–48]. Diagonal shoulder extension and flexion with theraband (ten repetitions each) was also practiced while participant sat on the ball. Level of resistance was adjusted individually by the attending physiotherapist. Participants continued to perform simple core stability and control exercise at home prescribed (wall squat, sitting

marching and diagonal flexion and extension in sitting on chair with theraband with each exercise repeated for ten times once/day), 3 times/week during the program period. At the post-program reassessment, all participants reported that they had neither taken any analgesics nor received other treatments apart from the rehabilitation program.

## Data processing and statistical analysis

The software Nexus (version 1.0, Vicon, Oxford, UK) was used for the kinematic and force plate data processing. Lumbar spine segment was determined between markers placed over the spinous process of L1 and the pelvis (defined by markers LASI, RASI, LPSI and RPSI). Hip kinematics was determined by the pelvis and markers of the lower limb over the ipsilateral side (defined by THI, KNE, TIB and ANK). The two phases of the bending cycle were determined by the upper limb movement for which the shoulder started to move until the fingers reach the floor or foam placed on the foam, vice versa for the extension phase. Three consecutive cycles (fourth to the sixth) were selected from the 7 cycles of bending for each trial. The kinematic data were then processed using a customized MATLAB code (Version R2016a, MathWorks Inc., Natick, MA, USA) in which the values of the angular velocity of the lumbar spine and hip joint were computed for analysis. The raw EMG signals were band-ass filtered between 10 and 500Hz, full-wave rectified, and low-pass filters with a cut-off frequency of 10Hz. The linear envelope of the EMG data was then normalized to the percentage of the MVC of the respective muscle, for further analysis of the pattern of recruitment.

Statistical analysis of the dependent variables was conducted using SPSS version 23 (SPSS, Chicago, USA); the alpha level was set at 0.05 for analysis. Normality of data was examined and verified by Shapiro–Wilk test. Paired t-test was used to compare the five clinical outcomes within the LBP group measured at the pre- and post-program assessment. One way ANOVA was used to compare the regional angular velocity and percentage of MVC for between-group (LBP vs healthy) and between-time (within LBP group) using post-hoc analysis (Scheffe test). Correlation analyses were performed using Pearson's correlation between the six clinical outcomes (i.e. SBTB (TOTAL), SBTB (SUB), TSK, PSFS, RMDQ and PSEQ, for assessing their associations before and after the program.

## Results and discussion

All participants were able to complete the testing protocol at pre-program and post-program assessments. Table 1 shows the demographics and clinical outcomes of the participants. There are no significant differences found between the healthy and LBP group in age, weight, height and body mass index (p>0.05). For comparisons of clinical outcomes collected in LBP group at pre- and post-program, significant differences are found in pain (VAS, p<0.001, Cohen's d = -4.9813); functional disability (PSFS, p<0.001, Cohen's d = 2.361); and self-efficacy (PSEQ, p = 0.046, Cohen's d = 0.6725). Changes in RMDQ score, SBTB (TOTAL and SUB) scores and TSK score are statistically insignificant for LBP participants (p>0.05). Comparisons of the hamstrings flexibility show no significant difference between-group or between-time (for pre- and post-program assessments in LBP group).

### Comparison of kinematic parameters at pre- and post-interventions

**Centre of pressure excursion.** Centre of pressure excursion (COP) measured in antero-posterior and medio-lateral directions is presented in Fig 2. All comparisons revealed insignificantly differences (p>0.05).

**Cycle time of bending and angular velocity of lumbo-pelvic kinematics.** Fig 3 shows the averaged value of the one cycle time in flexion and extension phases (in seconds) during

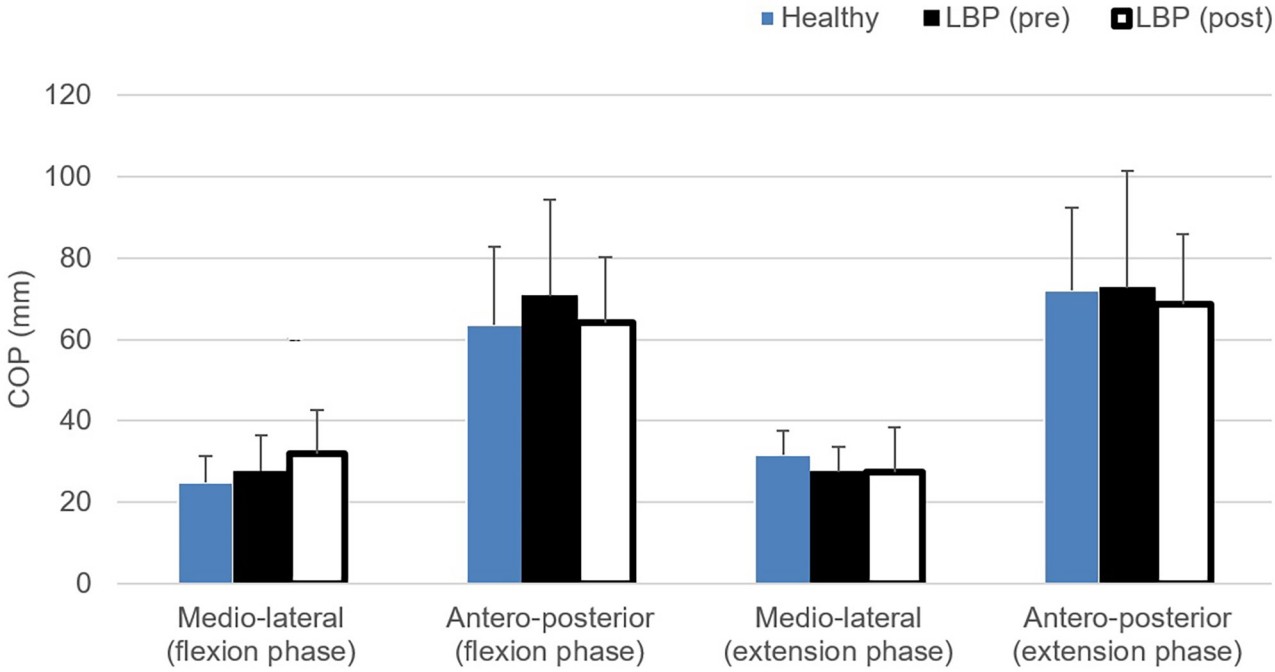

**Fig 2. Comparisons of the centre of pressure excursion (COP) during flexion and extension phase of the repeated bending task.**

the repeated bending test at pre- and post-intervention assessments. There is significant difference in the one cycle flexion time at baseline assessment before the healthy and LBP group (F = 9.505, p = 0.05, $\eta^2$ = 0.312). Fig 4 illustrates the regional angular velocity measured at the lumbar spine and hip joint during the flexion and extension phases of the bending test. Values of the lumbar spine and hip joint velocity in both flexion and extension were found to have significantly decreased in LBP group at baseline compared to the healthy group. Meanwhile, values of the regional velocity recorded at post-program assessment in LBP group were

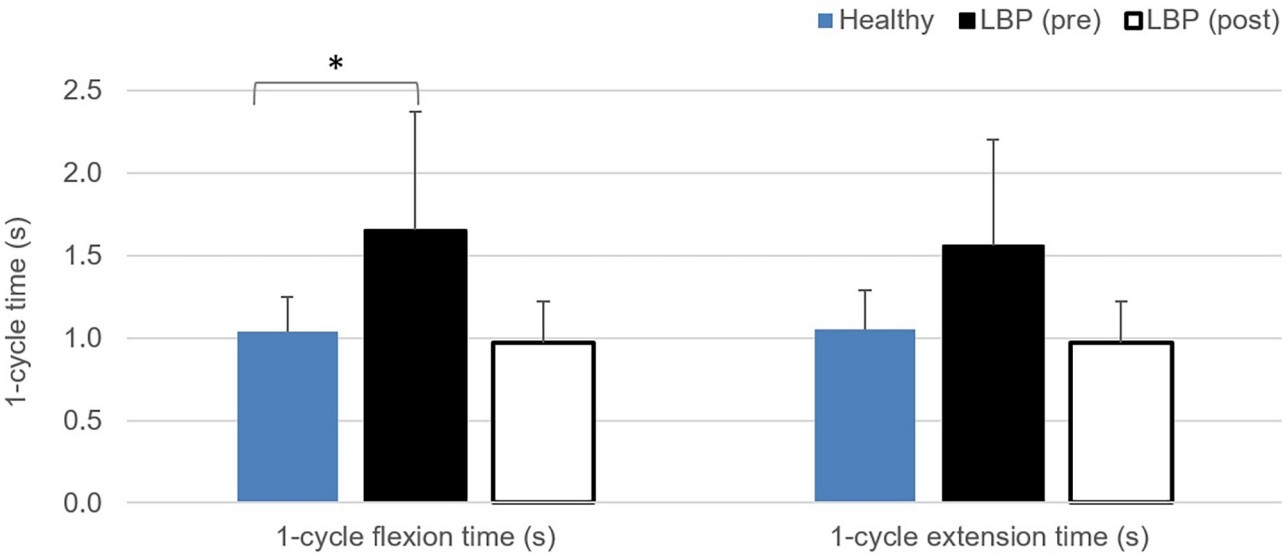

**Fig 3. Comparisons of the 1-cycle time (seconds) of the repeated bending task.** * indicates significant difference found between healthy and LBP (pre) with p<0.05.

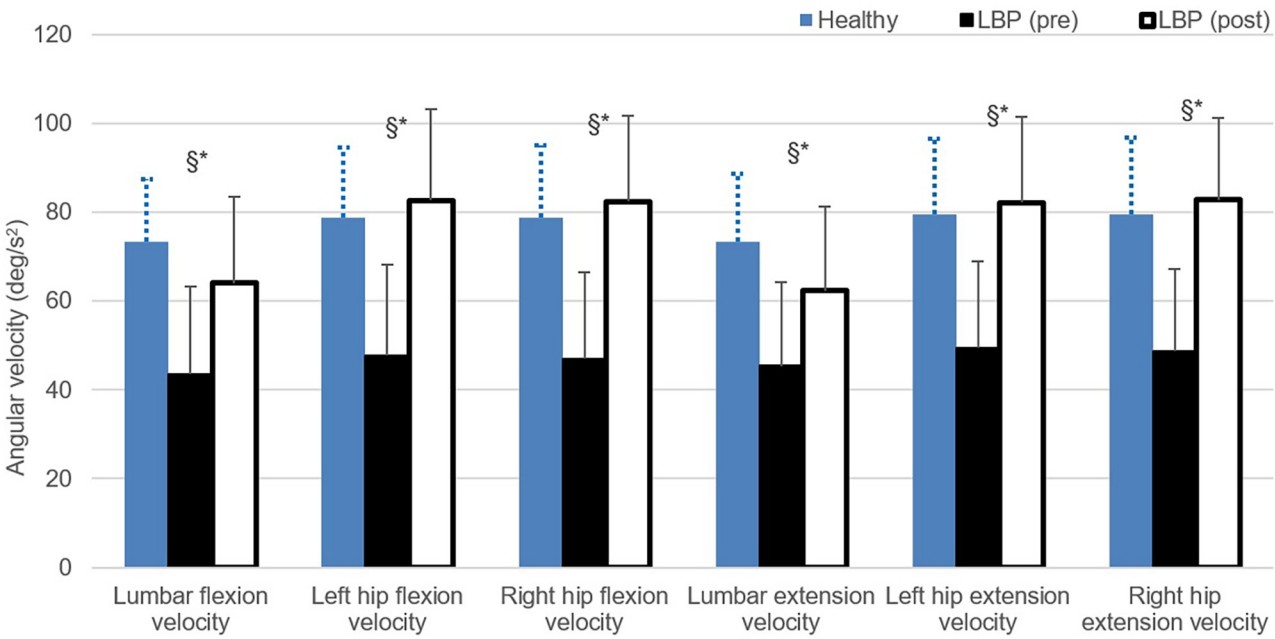

**Fig 4. Comparisons of lumbar spine and hip angular velocity during the repeated bending task.** * indicates significant difference found between healthy and LBP (pre); and § indicates significant difference found between LBP (pre) and LBP (post) with p<0.05.

significantly increased compared to the pre-program magnitude. No significant differences were found for the comparisons between the data collected at LBP (post) and healthy group.

## Muscle recruitment pattern

Figs 5 and 6 show the muscle recruitment pattern (expressed in % MVC) of the trunk and lower limb muscles during the repeated bending test, measured before and after the rehabilitation program for flexion and extension phases of the bending task respectively. For flexion phase of the bending, there were no significant differences found between healthy and LBP (pre) or between healthy and LBP (post). However, the pre-post comparisons of the data in the LBP group showed significantly increased values of % MVC for the right TFL (F = 2.514, p = 0.05, $\eta^2$ = 0.126), right LES (F = 3.084, p = 0.05, $\eta^2$ = 0.150), right BF (F = 3.933, p = 0.029, $\eta^2$ = 0.183) as well as bilateral MUL (left: F = 3.302, p = 0.049, η2 = 0.159; right: F = 3.999, p = 0.027, η2 = 0.186) and Gmax (left: F = 2.361, p<0.05, $\eta^2$ = 0.119; right: F = 1.424, p = 0.05, η2 = 0.075) during the flexion phase. For extension phase (recovery) of bending, there were no significant differences found between healthy and LBP (pre) group except the left BF in LBP group showed a significantly greater activity at pre-program assessment compared to the healthy group (F = 3.073, p = 0.05, $\eta^2$ = 0.149). The within-group analysis showed significantly greater activity level of right TFL (F = 3.642, p = 0.037, $\eta^2$ = 0.172) and left Gmax (F = 1.450, p = 0.05, $\eta^2$ = 0.077) for participants with LBP at post- compared to pre-program assessments. In addition, there are significant differences shown for comparisons between healthy and LBP (post) for EMG activity of the right TFL and left BF in which the LBP group activated these two muscles at a greater level compared the asymptomatic individuals.

## Correlations between clinical outcomes

Fig 7 shows the correlation matrix between the clinical outcomes (SBTS TOTAL and SUB, TSK, PSFS, RMDQ and PSEQ) of participants in LBP group, at pre- and post-program

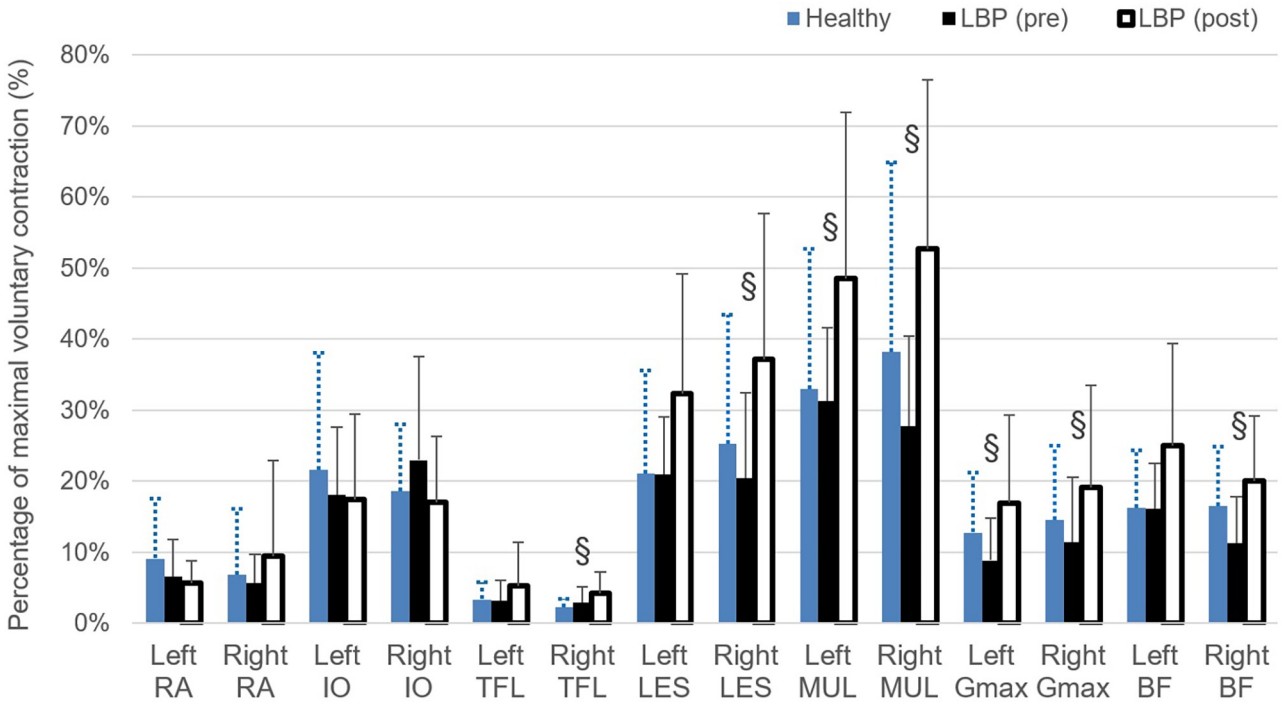

**Fig 5. Muscle recruitment (expressed in percentage of maximal voluntary contraction, MVC) during flexion phase of the bending task performed at self-preferred speed.** Significant difference was found between LBP (pre) and LBP (post) with p<0.05 (§). RA, rectus abdominus; IO, internal oblique; TEL, tensor fascia latae; LES, lumbar erector spinae; MUL, lumbar multifidus; Gmax, gluteal maximus; BF, biceps femoris.

analysis. At pre-program assessment, significant correlations were found between 1) SBTB (TOTAL) and SBTB (SUB) ($r = 0.918$, p<0.01), RMDQ ($r = 0.882$, p<0.01), PSFQ ($r = -0.671$, p<0.01); 2) between SBTS (SUB) and TSK ($r = 0.650$, p<0.05), PSFS ($r = -0.537$, p<0.05), RMDQ ($r = 0.882$, p<0.01), PSEQ ($r = -0.819$, p<0.01); 3) between TSK and PSEQ ($r = -0.545$, p<0.05); 4) between PSFS and PSEQ ($r = 0.561$, p<0.05); and 5) between RMDQ and PSEQ ($r = -0.758$, p<0.01). At post-program assessment, significant correlations were found in those reported for pre-program analysis (r values range from -0.790 to 0.949, p<0.05) except the associations between 1) SBTS (SUB) and TSK ($r = -0.220$, p<0.05), and 2) TSK and PSEQ ($r = -0.570$, p>0.05). One additional significant association was found at post-program between SBTB (TOTAL) and TSK ($r = 0.756$, p<0.01).

This study investigated the modification of lumbopelvic movements and motor control pattern during forward bending in individuals with chronic nonspecific LBP after completing a 6-week rehabilitation program. The comparative analysis of the physical outcomes between the pre- and post-program performance of the LBP group of participants to the healthy controls, reveals the degree and strategy of recovery of their dynamic movement control at the lumbo-pelvic region. This finding may have important clinical implications for the clinicians in understanding the patients' response to exercise training.

## Performance of LBP group at pre-program interval compared to healthy individuals

Participants in LBP group moved with significantly lower velocity at their lumbar spines and hip joint when executing the forward bending and recovery task, compared to the healthy

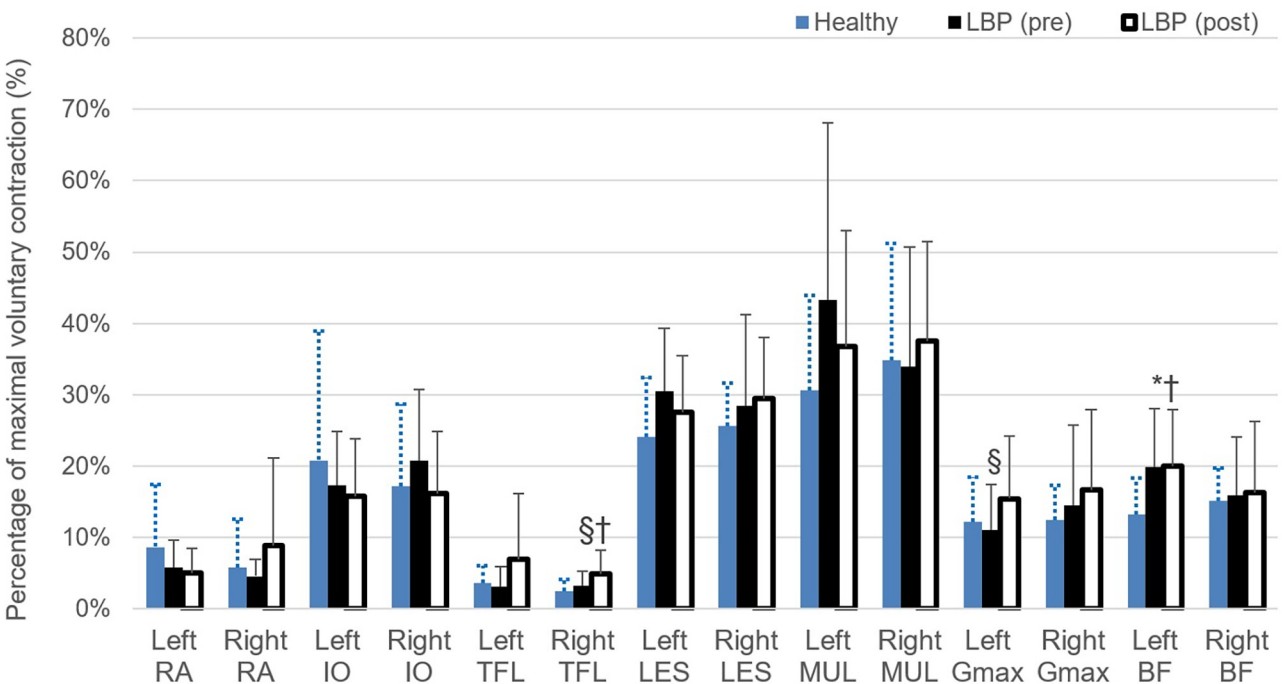

**Fig 6. Muscle recruitment (expressed in percentage of maximal voluntary contraction) during extension phase of forward bending task performed at self-preferred speed.** * indicates significant difference found between healthy and LBP (pre); § indicates significant difference found between LBP (pre) and LBP (post); and † indicates significant difference found between healthy and LBP (post). RA, rectus abdominus; IO, internal oblique; TEL, tensor fascia latae; LES, lumbar erector spinae; MUL, lumbar multifidus; Gmax, gluteal maximus; BF, biceps femoris.

controls at pre-program assessment. The compromised dynamic capacity of the lumbopelvic region found in this study is consistent with the impairment reported in the literature [46–49]. The reduced movement velocity could be partly explained by the higher degree of co-contraction of trunk flexor and extensor muscles in those with LBP while performing the trunk flexion task in a semi-seated position reported previously [47]. This particular muscle recruitment pattern substantiates the trunk bracing or stiffening strategy frequently adapted by the symptomatic group. Clinically, this phenomenon may explain the classical description given by those experiencing severe back pain: they say they feel a "locking" or "giving way" of the back when they try reaching or bending forward at a faster speed [46, 50]. The significantly lower movement velocity had been considered to be a protective and mal-adaptive movement pattern shown in LBP group potentially associated with their fear-avoidance belief towards movement [51]. Grotle et al. reported that patients with chronic LBP had significantly higher fear-avoidance belief (measured in Fear-Avoidance Beliefs Questionnaire, FABQ) than those with acute LBP [50]. More importantly, such difference remained unchanged at one-year follow-up. There is consistent evidence available to substantiate that level of fear avoidance predicts the intensity of pain and disability in LBP [52, 53]. Participants in this study showed the significant improvement with their confidence level in performing their daily activities in pain (evidenced by the significant increase in PSEQ) and close to significant reduction in the TSK score ($P$ = 0.053), after completing the rehabilitation program. Our findings indicate that both the aberrant movement pattern and fearfulness towards provocative movement could be optimised satisfactorily by re-education and training of the lumbopelvic region using specific therapeutic exercise regimen.

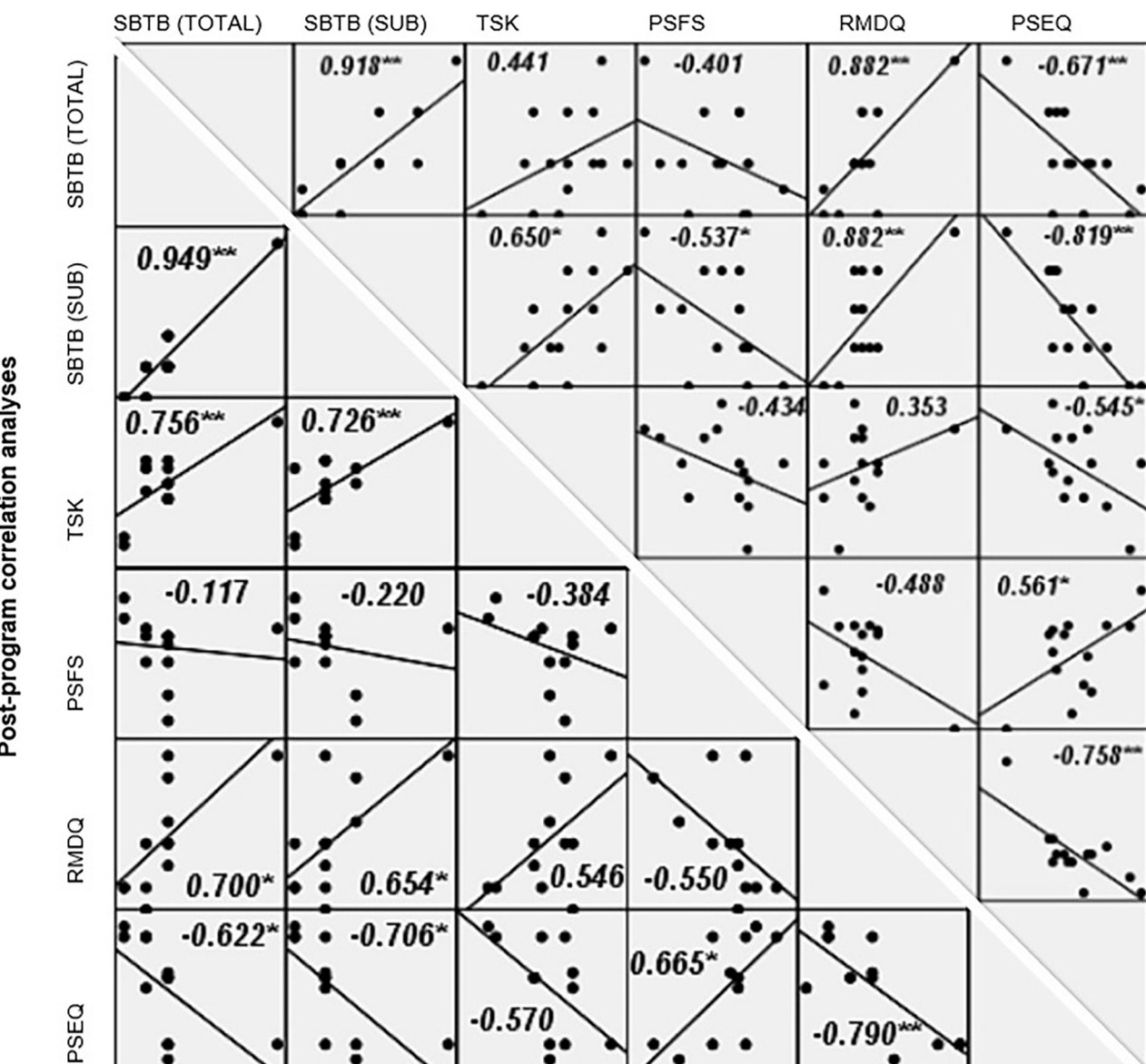

**Fig 7.** Correlation matrix between the clinical outcomes for participants in LBP group, at a) pre- and b) post-program interval (with correlation coefficient, r and p values [* <0.05 and ** <0.01] included). SBTB (TOTAL), STarT Back Tool (Total score); SBTB (SUB), STarT Back Tool (Sub-score); TSK, Tampa Scale of Kinesiophobia; PSFS, Patient Specific Functional Scale; RMDQ, Roland Morris Disability Questionnaires; and PSEQ, Pain Self Efficacy Questionnaire.

## Effects of exercise program on movement pattern and motor control strategies in LBP group

Marras et al. studied the movement-associated risk factors reported by industrial workers and their contribution to development of back pain [54, 55]. They reported that the probability of sustaining a back injury as a result of moving the trunk at high speed was double compared to that caused by moving the trunk at the maximum flexion angle. Participants in current LBP

group showed a satisfactory recovery of their lumbar spine movement velocity after the 6-week program which focused on optimizing the movement pattern between the lumbar spine and hip region. Our findings may suggest its potential application for reducing the injury or recurrence of the back condition. Despite the emerging evidence revealed in previous studies, which compared the rhythm of the lumbo-pelvic movements in people with LBP to healthy individuals, knowledge of impact of the speed level on the spinal movement strategy during forward bending remains limited. This knowledge gap indicates the possible pitfalls of assessing the functional activities at the self-preferred speed of individuals with LBP dysfunction since the condition itself could be self-limiting. Furthermore, the highly task-specific exercise training incorporated in this 6-week program might also have promoted the feedforward mechanisms, a crucial pathway that monitor movement and functional organization of the cortex associated with musculoskeletal dysfunctions [56–58]. Previous neurophysiological studies which examined the underlying mechanisms of the effect of task-specific training on the recovery of motor control and management of the musculoskeletal dysfunction revealed that exercise training with task specificity is crucial for neural and functional rehabilitation [56]. To ensure the safety and suitability of the patients to participate in the speed dependent task or training program, proper procedures for screening of the contraindications and identification of subgroup of patients with LBP are recommended.

This is the first time to reveal that individuals with chronic LBP manifested with compromised movement capacity at the recurrent or exacerbation phase could recover to the level of performance comparable to the healthy individuals in 6 weeks as substantiated by the lack of between-groups differences in the kinematics data collected at LBP (post-program) and healthy group. Furthermore, it is crucial to note that positive reorganization of muscle recruitment was revealed in the LBP group with the significant enhanced activity level of lumbar multifidus and gluteus maximus bilaterally during the flexion phase of the bending cycle, after the program [59, 60]. The improved level of willingness and capability of the symptomatic group to execute this "fearful or provocative" functional movement at a faster pace implies promising impact of the exercise training in terms of the interplay between the physical and psychological components involved in chronic pain management in spine dysfunction [61, 62]. Hypermobility may present in the spinal segments or regions of those with LBP with diminished resistance to segmental manual displacement applied to the spine which contributes to the typical manifestations of spinal instability [61–63]. In a recent review of the evolution of the concepts of stability and instability relevant to back pain by Revees [63], sufficient spinal stiffness is required to support the upper body part during dynamic activities and this is achieved by the fine-tuning and timely activation of the spinal muscles. These mechanisms are critical for protecting the spine from injury and pain for its potential role of modified preparatory trunk control prior to the predictable spine perturbation [64].

Steiger et al. [65] reported in their systematic review that evidence was lacking to support the notion of treatment effects of exercise therapy in chronic LBP directly attributable to changes in the musculoskeletal system as displayed in trunk muscle strength and endurance. Contrary to their results, our positive impacts on the clinical and physical outcomes indicate that the study of the motor control strategy using movement velocity analysis and recruitment of the specific lumbopelvic muscles region would be more sensitive and specific to the induced changes occurring at the neuromusculoskeletal system for this subgroup of chronic LBP. This is substantiated by the recent clinical commentary by Hodges et al. [66] which reiterates the importance in considering the task-specific changes in the multifidus and erector spinae of the lumbar spine as a result of injury and recovery. The modification of the muscle recruitment pattern revealed in the present study provide novel evidence to support the use of motor control training to first overcome the adaptive muscle inhibition during the acute exacerbation

phase [67], and second to facilitate the restoration of the optimal recruitment pattern during the rehabilitation phase [60, 68].

COP excursion is commonly used to measure the balance performance and postural stability in nonspecific LBP. Previous research reviewed that when compared to healthy control, individuals with LBP displayed a significantly greater COP excursion consistently across studies in antero-posterior (range: 2.3–7.5mm) and in medio-lateral direction (range: 1.6–4.7mm) when performing the normal stance task [69]. In this study, we showed that the COP excursions in the antero-posterior direction were similar between the LBP and healthy group following the 6-week program, suggesting an improvement in postural stability in the LBP group. Numerous factors, for examples, the presence of pain, spine mobility capacity and muscular excitation affect the postural stability in individuals with LBP [70]. This study also showed substantial increase in the bending pace in LBP group and the same factors may have also contributed to the improvements in spine movement velocity [71]. It has been proposed that postural sway increase in low back pain is not related to a reduced spinal range of motion, but the increase in muscular active tension, which reduces dynamic mobility capacity. Based on the present findings, pain intensity, spine movement velocity and recruitment of the key spinal muscles all showed a positive recovery upon the completion of the 6-week structured rehabilitation program. It therefore suggests that this specific program plays a critical role in optimising the neuromusculoskeletal dysfunctions that underlie this cohort of nonspecific LBP.

## Associations between clinical outcomes

The role and negative impact of fear-avoidance behaviour and self-efficacy associated with the progress and prognosis of LBP had been reported previously [72–74]. Movement strategies relevant to the LBP subgroup with dominant psychosocial contributors include bracing of the trunk manifested in reduced dissociation of rotation between the thoracic spine and pelvis during walking [75–77]; enhanced stiffness of the trunks and hip joints to minimise the internal perturbation during repeated bending [15, 56, 57]. Meanwhile, slower movement pace displayed in LBP group could be considered as an effective strategy adopted by the individuals to avoid pain provocation; however, it would be considered mal-adaptive if this pattern persists beyond the recovery phase. Both the fear-avoidance and self-efficacy scores improved in our LBP group after the program. The present findings suggest that the significant recovery of pain (by VAS) and function (by PSFS) in the symptomatic group could have been partly mediated by modifying their kinesiophobia and avoidance hypervigilance, as well as empowering patients with LBP to confidently engage in their daily functions. It had been found that self-efficacy is more important than fear of movement in mediating the relationship between pain and disability in chronic LBP population [78].

The present findings were obtained from a small cohort of relatively young adults suffered from chronic nonspecific LBP. Further research is required to study the applicability of the exercise program and the generalisation of these findings to other age groups. It is recommended to carry out randomized controlled studies with large sample size and longer-term follow-up period to truly reveal the clinical course and recovery of individuals with nonspecific LBP in future for the better the clinical management and preventive measures for this highly disabling subgroup of spine dysfunction.

## Conclusions

Individuals with chronic nonspecific LBP moved with a strategy featured with a significantly slower movement speed both at their lumbar spine and hip articulation compared to the able-bodies when performing trunk forward bending in standing. Such movement strategy may

indicate the suboptimal efficacy of the dynamic muscle system exhibited in the symptomatic group. Upon completion of a structured program which emphasizes on re-education and training of the lumbopelvic movement control, individuals with LBP showed a satisfactory recovery of their movement speed and reorganization of the muscle activation patterns with the enhanced activation level of the local stabilizer muscles of the lumbopelvic region evidenced during bending task. More profound reorganization of the motor control was found during the flexion phase than the extension phase of the bending task. These findings indicate that the recovery of the movement and motor control pattern in patients with chronic LBP achieved to a comparable level of the healthy able-bodies. The improvement of both the physical outcome measures suggest that specific rehabilitation program would help promoting the functional recovery of this specific subgroup of LBP.

## Supporting information

**S1 File. Questionnaires and scales.**
(PDF)

## Acknowledgments

The authors would like to thank the participants of this study, Ms Veronica Liu for conducting the 6-week program at the Department of Physiotherapy at Prince of Wales Hospital; Ms Donna Tam and Ms Phoebe Xie for assisting the data collection; Mr. Jay Dai for preparing the customised MATLAB code for data analysis.

## Author Contributions

**Conceptualization:** Sharon M. H. Tsang, Grace P. Y. Szeto, Raymond Y. W. Lee.

**Data curation:** Sharon M. H. Tsang, Angelina K. C. Yeung, Eva Y. W. Chun.

**Formal analysis:** Sharon M. H. Tsang, Grace P. Y. Szeto, Caroline N. C. Wong, Raymond Y. W. Lee.

**Funding acquisition:** Sharon M. H. Tsang.

**Investigation:** Sharon M. H. Tsang, Eva Y. W. Chun.

**Methodology:** Sharon M. H. Tsang, Grace P. Y. Szeto, Angelina K. C. Yeung, Caroline N. C. Wong, Edwin C. M. Wu, Raymond Y. W. Lee.

**Project administration:** Sharon M. H. Tsang, Angelina K. C. Yeung, Eva Y. W. Chun, Edwin C. M. Wu.

**Resources:** Sharon M. H. Tsang, Angelina K. C. Yeung, Eva Y. W. Chun, Caroline N. C. Wong, Edwin C. M. Wu.

**Writing – original draft:** Sharon M. H. Tsang, Raymond Y. W. Lee.

**Writing – review & editing:** Sharon M. H. Tsang, Grace P. Y. Szeto, Angelina K. C. Yeung, Eva Y. W. Chun, Caroline N. C. Wong, Edwin C. M. Wu, Raymond Y. W. Lee.

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
