## [Decision Letter · Decision Letter 0]

29 Jun 2021

PONE-D-21-03559

Recovery of the lumbopelvic movement and muscle recruitment patterns using motor control exercise program in people with chronic nonspecific low back pain: a prospective study

PLOS ONE

Dear Dr. Tsang,

Thank you for submitting your manuscript to PLOS ONE. After careful consideration, we feel that it has merit but does not fully meet PLOS ONE’s publication criteria as it currently stands. Therefore, we invite you to submit a revised version of the manuscript that addresses the points raised during the review process.

ACADEMIC EDITOR:

Two expert in the fields revise your ms advising some major issues. Please consider also the indications in the revision process.

We look forward to receiving your revised manuscript.

Kind regards,

Emiliano Cè

Academic Editor

PLOS ONE

Journal Requirements:

2.  Please amend your Methods section to include the name of the hospital or clinic where patients were recruited. Additionally, we ask that you provide a copy of the questionnaire used, as a supplemental file. "

For additional information about PLOS ONE ethical requirements for human subjects research, please refer to " ext-link-type="uri" xlink:type="simple">http://journals.plos.org/plosone/s/submission-guidelines#loc-human-subjects-research."

4. Please ensure that you include a title page within your main document. You should list all authors and all affiliations as per our author instructions and clearly indicate the corresponding author.

5. Please upload a copy of Supporting Information Figures and Tables which you refer to in your text.

6. We note that Figure 1 includes an image of a participant in the study. 

Reviewers' comments:

Reviewer's Responses to Questions

**Comments to the Author**

1. Is the manuscript technically sound, and do the data support the conclusions?

Reviewer #1: Partly

Reviewer #2: Partly

2. Has the statistical analysis been performed appropriately and rigorously? 

Reviewer #1: I Don't Know

Reviewer #2: N/A

3. Have the authors made all data underlying the findings in their manuscript fully available?

Reviewer #1: No

Reviewer #2: Yes

4. Is the manuscript presented in an intelligible fashion and written in standard English?

Reviewer #1: Yes

Reviewer #2: Yes

5. Review Comments to the Author

Reviewer #1: Your paper seems to be interesting. Please find some questions:

1. How the sample size was set? 15 patients with chronic LBP seems to be a small group for chronic LBP

2. Were groups included women and men?

3. The mean age of participants was about 30. Is it a typical age for an onset of LBP? May the results be extrapolated for other age groups?

4. How the COP excursions were measured? Only with markers on the 2nd metatarsals? Is this methodology standard in COP measurements?

5. How core muscles activation (eg. transversus abdominis) was achieved in patients?

6. How many repetitions of exercises were performed?

7. How would you explain greater excursion in COP in LBP in posttreatment?

127/128 The linear envelope of the EMG data was then normalization to the percentage of the MVC of the respective muscle, for further analysis of the pattern of recruitment.

132 the program

180 Significant difference was found.

Reviewer #2: Tsang et al. Plos One – Recovery of the lumbopelvic movement and muscle recruitment patterns using motor control exercise program in people with chronic nonspecific low back pain: a prospective study

Overview: The present study describes the impact on a 6-week rehabilitation program on lumbopelvic and muscle recruitment pattern in a population affected by chronic nonspecific low back pain. I think that this work illustrates an argument very interesting, especially in a practical point of view: it concerns one of the most common disease which limits the daily activity. However, there are some points that need some clarification and revision. See my specific comments below.

Introduction:

Line 39: a brief definition of movement-based intervention should be provided.

Line 40: Please, clarify “interventions in the lumbo-pelvic kinematics”.

A brief paragraph about the impact and the role of motor control on chronic low back pain should be add. It is only mentioned in the line 26.

Moreover, this part should be extended with the reference about the impact of different type of exercise, rehabilitation programs or training in this type of population (van Middelkoop M et al., 2010; Searle et al., 2015; Anderson et al., 2017).

Materials and methods:

In the participant paragraph, I suggest specifying their demographic characteristics or a mention of table 1.

Line 57: The authors should add the year of Declaration of Helsinki mentioned.

Were asked to participant to refrain from any heavy physical activity of lower limbs in the 48 h prior to the study?

In “Clinical outcome assessments” paragraph, I suggest the authors providing a brief explanation of the purpose of the single questionnaires proposed.

Physical outcomes assessments

Line 71: Was a warm-up be performed before the 3-5 bending trials? And if so, was it standardized for each group and participant? In that case, specify what the warm-up trials consisted in

Was the test conducted at the same time of the day for the reliability of the measurement? (Ensink 1996).

Line 86: Please, add a reference for the standardized skin preparation.

Lines 87-89: I suggest the authors to briefly explain the muscles chosen and to support it with references.

Considering the “Surface electromyography” paragraph, it lacks the material used, specifying the name and the country of manufacturer.

Regarding the kinematics acquisition, It should be added the recording frequency and the number of camera which constitute the optoelectronic motion capture system.

6-week structured rehabilitation program

Please, provide further information about the rehabilitation program (i.e. number of series, number of repetitions, seconds..) as the exercise performed at home

Statistical analysis

Was the normality tested? Was also used a post hoc test?

Results

Please, in table 1 description specify how the results were shown (i.e. mean±standard deviation (SD))

Table 1: TSK and PSEQ were not declared.

I recommend the authors to give the partial eta squared values

In the graphs I the authors to show the healthy population’s results with a further bar (as figure 2 and 3). The line suggests a progression and it is not clear.

Discussion:

Line 212: Please clarify the sentence; “Participants in LBP group moved with a significantly lower lumbar spine”.

Lines 242-243: Please, review this sentence

It lacks the centre of pressure excursion variation explanation

6. PLOS authors have the option to publish the peer review history of their article (what does this mean?). If published, this will include your full peer review and any attached files.

Reviewer #1: No

Reviewer #2: No

---

## [Author Response · Author response to Decision Letter 0]

31 Aug 2021

PONE-D-21-03559R1

Recovery of the lumbopelvic movement and muscle recruitment patterns using motor control exercise program in people with chronic nonspecific low back pain: a prospective study

PLOS ONE

Response to Review Comments 

Reviewer #1: Your paper seems to be interesting. Please find some questions:

1. How the sample size was set? 15 patients with chronic LBP seems to be a small group for chronic LBP.

Response: The sample size for LBP group was calculated to range between six and twenty participants based on the effect size of the clinical outcomes (Visual Analogue Scale (VAS), Patient-Specific Functional Scale (PSFS), Roland Morris Disability Questionnaire (RMQ), and Patient Self Efficacy Questionnaire (PSEQ). The authors agreed that the current sample size is comparatively small for study of LBP. Although there were much greater number of participants underwent this 6-week program at the clinical centre, due to the complexity and time demand to conduct the biomechanical outcomes before and after the 6-week intervention program, not many of the program participants voluntarily consented to take part in the biomechanical assessments and reassessments. 

2. Were groups included women and men?

Response: There were 8 females and 7 males recruited each in the LBP group and asymptomatic group, and this information has been added (line 56 and in Table 1).

3. The mean age of participants was about 30. Is it a typical age for an onset of LBP? May the results be extrapolated for other age groups?

Response: According to the epidemiological data of the Institute of Health Metrics and Evaluation (IHME), the prevalence of LBP increases and peaks between the ages of 35 and 55. With the mean age of the present cohort of LBP participants of 34.1 i.e. close to the minimal age range, care would be needed when generalising the findings to other age groups. This limitation has been added in last paragraph of the discussion section (line 308-312). 

4. How the COP excursions were measured? Only with markers on the 2nd metatarsals? Is this methodology standard in COP measurements?

Response: The excursions of the Centre of Pressure (COP) in antero-posterior and medio-lateral directions were measured by the force plate (PF5000, AMTI, Massachusetts, USA) at 100 Hz where the feet were positioned (see Fig 1) and synchronised with the data collection of the kinematics by motion capturing system (Vicon) when performing the repeated bending tasks. The force plate data was analysed using the Nexus software of the Vicon system. The reporting of the COP measurement with the reference of the methodology standard of COP measurement has been added in the method section (line 113-115; �44 MacRae et al., 2018). 

5. How core muscles activation (e.g. transversus abdominis) was achieved in patients?

Response: The ability and accuracy of the core muscle activation of each participant in the LBP group was first assessed and corrected by the responsible physiotherapist using the manual palpation of the muscle tone and in-drawing manoeuvre of the anterior abdominal muscles just at 2cm medial to the anterior superior iliac crest of both sides (line 123-125; �45 Hides et al., 2000).

6. How many repetitions of exercises were performed?

Response: The details of the rehabilitation program and home exercise program in terms of the repetition and frequency have been added to the method section (line 126-131).

7. How would you explain greater excursion in COP in LBP in post-treatment?

Response: Authors have carefully gone through the analysis again and rectified that there was no significant difference found in all comparisons of the COP excursion between healthy group and LBP at pre- and post-program assessment. The reporting of the result has been rectified (line 166-169, 502 and Fig 3). In addition, some discussion of the insignificant between-group (healthy vs LBP group) and within-group (LBP) of the centre of pressure excursion as an index of postural stability has been added to the discussion section (line 284-294).

8. 127/128 The linear envelope of the EMG data was then normalization to the percentage of the MVC of the respective muscle, for further analysis of the pattern of recruitment, 132 the program and 180 Significant difference was found. 

Response: These grammatical errors have been revised (line 142 and 200).

 

Reviewer #2:

Overview: The present study describes the impact on a 6-week rehabilitation program on lumbopelvic and muscle recruitment pattern in a population affected by chronic nonspecific low back pain. I think that this work illustrates an argument very interesting, especially in a practical point of view: it concerns one of the most common disease which limits the daily activity. However, there are some points that need some clarification and revision. See my specific comments below.

Introduction:

1. Line 39: a brief definition of movement-based intervention should be provided.

Response: A brief definition of movement-based intervention has been added to the Introduction section to strengthen the objective and hypotheses of this study (line 43-45, references �25 to �27). 

2. Line 40: Please, clarify “interventions in the lumbo-pelvic kinematics”.

A brief paragraph about the impact and the role of motor control on chronic low back pain should be add. It is only mentioned in the line 26. Moreover, this part should be extended with the reference about the impact of different type of exercise, rehabilitation programs or training in this type of population (van Middelkoop M et al., 2010; Searle et al., 2015; Anderson et al., 2017).

Response: The comparative benefits of different types of exercise therapy for chronic low back pain and more details about movement-based interventions have been added in the introduction section respectively (line 39-42 and 43-45). New references have been included in the manuscript (�22 to �24 and �25 to �27).

Materials and methods:

3. In the participant paragraph, I suggest specifying their demographic characteristics or a mention of table 1.

Response: The mentioning of Table 1 which contains the demographic characteristics of the participants has been added to the participant paragraph (line 56). 

4. Line 57: The authors should add the year of Declaration of Helsinki mentioned.

Response: The year of the Declaration of Helsinki has been added (line 62).

 

5. Were asked to participant to refrain from any heavy physical activity of lower limbs in the 48 h prior to the study?

Response: Yes, all participants in this study (both healthy group and LBP group) were refrained from performing heavy physical activity of their lower limbs 2 days before the assessment and reassessment sessions (line 79-80). 

6. In “Clinical outcome assessments” paragraph, I suggest the authors providing a brief explanation of the purpose of the single questionnaires proposed.

Response: The introductory sentence has been revised to briefly explain the purpose of the corresponding questionnaire or scale used for quantifying the clinical outcomes of this study (line 66-68). 

7. Physical outcomes assessments

Line 71: Was a warm-up be performed before the 3-5 bending trials? And if so, was it standardized for each group and participant? In that case, specify what the warm-up trials consisted in.

Response: The baseline assessment and reassessment conducted before and after the 6-week program were carried out on a separate day of the program. Participants were allowed to practise the bending task according to the standardised instructions 3-5 times (line 78-79) so as to get themselves familiar with the task. There was no additional warm-up included for the preparation of the participants for these two assessment sessions. 

8. Was the test conducted at the same time of the day for the reliability of the measurement? (Ensink 1996).

Response: The assessment and reassessment of the individual participant in the LBP group were conducted at the similar time of the day because a specific period of the day (10am to 12noon) has been assigned for the assessment procedure according to the schedule of the clinical centre. 

9. Line 86: Please, add a reference for the standardized skin preparation.

Response: The reference for the standardized skin preparation has been added (�35 at line 95). 

10. Lines 87-89: I suggest the authors to briefly explain the muscles chosen and to support it with references.

Response: The rationale for the selected muscle groups under examination has been added along with the references �33 and �34 (line 92-94). 

11. Considering the “Surface electromyography” paragraph, it lacks the material used, specifying the name and the country of manufacturer.

Response: The name and the country of the manufacturer of the electromyography system used in this study has been added (line 89). 

12. Regarding the kinematics acquisition, it should be added the recording frequency and the number of camera which constitute the optoelectronic motion capture system.

Response: A total of ten cameras were used for acquisition of the kinematics data (information added to line 105) and the recording frequency (sampling frequency) was included (line 111).

13. 6-week structured rehabilitation program

Please, provide further information about the rehabilitation program (i.e. number of series, number of repetitions, seconds..) as the exercise performed at home.

Response: The details of the rehabilitation program and home exercise program in terms of the repetition and frequency have been added to the method section (line 120-130).

14. Statistical analysis

Was the normality tested? Was also used a post hoc test?

Response: Yes, normality of the data was screened and verified by Shapiro-Wilk test (line 145). Paired t-test was used to compare the five clinical outcomes within the LBP group measured at the pre- and post-program assessment (line 145-146). The comparison of the kinematics and electromyographic data for between-group (low back pain vs healthy) and between-time (within low back pain group) was conducted using the one way ANVOA and post-hoc analysis (Scheffe test) was used (line 147-149). 

Results

15. Please, in table 1 description specify how the results were shown (i.e. mean±standard deviation (SD)).

Response: The presentation format of the mean and standard deviation in Table 1 has been specified (line 159).

16. Table 1: TSK and PSEQ were not declared.

Response: The full term of PSEQ (Pain Self Efficacy Questionnaire) and TSK (Tampa Scale of Kinesiophobia) has been declared in the legend of Table 1 (line 160-161). 

17. I recommend the authors to give the partial eta squared values.

Response: The effect sizes expressed in Cohen’s d value for the clinical outcomes, namely the Visual Analogue Scale, Patient Specific Functional Scale and Pain Self Efficacy Questionnaire by comparing the pre- and post-program data have been added (line155-156). Meanwhile, the partial eta squared values for the comparison of the kinematics and electromyography between the healthy controls, LBP at pre- and post-program have also been added in the result section (line 174, 189-195).

18. In the graphs I the authors to show the healthy population’s results with a further bar (as figure 2 and 3). The line suggests a progression and it is not clear.

Response: The results of the healthy population are now illustrated with the bar in Fig 4 to 6, as recommended.

Discussion:

19. Line 212: Please clarify the sentence; “Participants in LBP group moved with a significantly lower lumbar spine”.

Response: This sentence has been revised to improve the clarity, “Participants in LBP group moved with significantly lower velocity at their lumbar spines and hip joint when …” (line 229).

20. Lines 242-243: Please, review this sentence

Response: This sentence has been revised as “To ensure the safety and suitability of the patients to participate in the speed dependent task or training, proper procedures for screening of the contraindications and identification of subgroup of patients with LBP are recommended.” (line 259-260)

21. It lacks the centre of pressure excursion variation explanation

Response: Authors have carefully gone through the analysis again and rectified that there was no significant difference found in all comparisons of the COP excursion between healthy group and LBP at pre- and post-program assessment. The reporting of the result has been rectified (line 165-167, 502 and Fig 3). Some discussion of the insignificant between-group (healthy vs LBP group) and within-group (LBP) of the centre of pressure excursion as an index of postural stability has been added to the discussion section (line 284-294).

End of response letter, thank you for reviewing our revised manuscript.

---

## [Decision Letter · Decision Letter 1]

20 Oct 2021

Recovery of the lumbopelvic movement and muscle recruitment patterns using motor control exercise program in people with chronic nonspecific low back pain: a prospective study

PONE-D-21-03559R1

Dear Dr. Tsang,

We’re pleased to inform you that your manuscript has been judged scientifically suitable for publication and will be formally accepted for publication once it meets all outstanding technical requirements.

Kind regards,

Emiliano Cè

Academic Editor

PLOS ONE

Additional Editor Comments (optional):

Reviewers' comments:

Reviewer's Responses to Questions

**Comments to the Author**

1. If the authors have adequately addressed your comments raised in a previous round of review and you feel that this manuscript is now acceptable for publication, you may indicate that here to bypass the “Comments to the Author” section, enter your conflict of interest statement in the “Confidential to Editor” section, and submit your "Accept" recommendation.

Reviewer #1: All comments have been addressed

Reviewer #2: All comments have been addressed

2. Is the manuscript technically sound, and do the data support the conclusions?

Reviewer #1: Yes

Reviewer #2: Yes

3. Has the statistical analysis been performed appropriately and rigorously? 

Reviewer #1: Yes

Reviewer #2: Yes

4. Have the authors made all data underlying the findings in their manuscript fully available?

Reviewer #1: Yes

Reviewer #2: Yes

5. Is the manuscript presented in an intelligible fashion and written in standard English?

Reviewer #1: Yes

Reviewer #2: Yes

6. Review Comments to the Author

Reviewer #1: Thank you for revising the manuscript. All questions have been responded. I am satisfied with the corrections.

Reviewer #2: I congratulate you on the paper that is now complete and fluent. I appreciate all the corrections and elaborations made by the authors. There are below my further few little suggestions:

I suggest replacing the year of Helsinki Declaration with 2013. Moreover, I reccomend to separate result and discussion title. Please, add in the text the information about the time of a day of the measurement, as the authors did in the reviwer's comment.

7. PLOS authors have the option to publish the peer review history of their article (what does this mean?). If published, this will include your full peer review and any attached files.

Reviewer #1: No

Reviewer #2: **Yes: **Marta Borrelli

---

## [Editor Report · Acceptance letter]

27 Oct 2021

PONE-D-21-03559R1 

Recovery of the lumbopelvic movement and muscle recruitment patterns using motor control exercise program in people with chronic nonspecific low back pain: a prospective study 

Dear Dr. Tsang:

I'm pleased to inform you that your manuscript has been deemed suitable for publication in PLOS ONE. Congratulations! Your manuscript is now with our production department. 

Kind regards, 

on behalf of

Professor Emiliano Cè 

Academic Editor

PLOS ONE